# Mass concentration intercomparison of soot generated with Mini-Cast

Benoît Sagot<sup>1</sup>, Guillaume Pailloux<sup>2</sup>, Amel Kort<sup>2</sup>

<sup>1</sup> ESTACA, ESTACA'Lab - Paris Saclay, Montigny-Le-Bretonneux, F-78180, France <sup>2</sup>Autorité de Sûreté Nucléaire et de Radioprotection (ASNR), PSN-RES/SCA/LPMA, F-91400, Saclay, France

Correspondence to: Benoît Sagot (Benoit.sagot@estaca.fr)

Abstract. This study focuses on measuring mass concentration of soot aggregates generated with a Mini-CAST burner. The experiments were performed in a test bench able to generate soot particles with different size distributions and different OC/TC ratios. With this soot production, we assessed the mass concentration measurements based on three online instruments (TEOM, PPS and MA300) and two offline concentration determination (OC/TC and SMPS), considering the gravimetric measurement

- as a reference. The findings demonstrate that the TEOM and the quantification based on the thermo-optical OC/TC analyser performed within acceptable limits of 10 % in comparison to the gravimetric reference measurement, over a wide range of OC/TC, mass concentration and size distribution. The Pegasor Particle Sizer (PPS) mass concentration measurement which is based on the aerosol electrical charging is calibrated for a reference size distribution, and we suggested a correction of the mass concentration measurement based on the aerosol Fuchs active surface, that proved to be efficient within the limits of this
- study. Finally, we confirmed that the mass concentration measurements obtained with the MA300 aethalometer are OC/TC ratio and wavelength dependent, and we were able to establish OC/TC limits for the overall mass concentration evaluation with the infrared and ultraviolet wavelengths.

# **1** Introduction

In the context of studies on airborne dispersion of soot particles emitted during fire scenarios and monitoring and quantification of emissions from thermal engine, it is important to have a robust measurement of the mass concentration of emitted soot. These measurements are needed to assess the consequences of fires in Basic Nuclear Installations (Kort et al., 2022), as soot is responsible for clogging the last containment barrier made by HEPA filters, or for the metrology of conventional engines PM emissions (Aakko-Saksa et al., 2022). Dynamic real-time monitoring of mass emissions can be performed using various types of instruments based on different measurement principles, such as the real-time aethalometer-type measurement

- (MA300), the Tapered Element Oscillating Microbalances (TEOM) or the Pegasor Particle Sensor (PPS). These instruments allow the study of aerosols across a wide range of concentrations, from ambient air quality monitoring to highly concentrated exhaust emissions. The quantification performed by the PPS is based on the electrical charge on the surface of aerosols, making it extremely dynamic. However, it requires calibration that primarily depends on the median diameter of the aerosol particle size distribution. Similarly, the aethalometer is an instrument based on optical transmission through a soot cake collected on a
- filter, and its mass quantification is influenced by the physical nature of the soot, particularly the elemental carbon to organic

carbon ratio. For time stable emission sources, it is also possible to perform measurements using filter sampling, which allows mass concentration quantification either by weighing or through the Sunset Laboratory's OC/TC (Organic Carbon to Total Carbon ratio) analyser. All these mass concentration measurement techniques, usually used in different contexts, need to be qualified in an intercomparison to provide reliable and robust measurements.

- For this purpose, soot was generated by a mini-CAST JING 5201C burner through the combustion of propane. Different studies have demonstrated that this mini-CAST generator can produce soot with a wide range of sizes and properties and is considered as a relevant tool to produce soot similar to those emitted by different combustion processes such as aircraft engine (Marhaba et al., 2019) or fire situation (Kort et al., 2021). It has been extensively utilized for soot characterization and as a time stable soot production source: by varying the gas flow rates supplied to this generator, it is possible to adjust the mass concentration
- over a wide range, as well as the particle size distribution and the elemental carbon to organic carbon ratio of the produced soot particles. The concentrations of soot generated are then either collected on a filter for ex-situ measurements or diluted using a double stage ejector-type diluter Dekati E-diluter, which delivers concentrations compatible with the measurement ranges of the instruments used.

The mass concentration evaluated by these online devices (MA300, PPS Pegasor, TEOM) were studied as a function of the 45 mass concentration evaluated by a direct gravimetric measurement and considered as the reference measurement. The evaluation of the OC/TC ratio and the size distributions of the produced soot particles are also necessary for the interpretation

of the instrument responses. A Scanning Mobility Particle Sizer (SMPS) was thus associated to the online instruments.

## 2 Experimental setup

For this experimental study we developed a test bench (Fig. 1) dedicated to the mass concentration intercomparison. This setup 50 includes the soot generation source, the mini-CAST, followed by two heated lines maintained at 180°C. One line is used for filter sampling, while the other feeds a dilution system that distributes the diluted and cooled aerosol to various measurement instruments: the TEOM, PPS, MA300, and SMPS. The following sections detail the operating principles and implementation conditions of each component of this test setup.

#### 55 Figure 1 – Configuration of the experimental bench for intercomparison of mass concentration measurements

The aerosol source is a Mini-CAST 5201C soot generator, which produces polydisperse soot particles by propane combustion. The principle of soot generation by the Mini-CAST involves an axial laminar diffusion flame, which is quenched by a cold, dry nitrogen flow to stop the combustion process and prevent soot oxidation. A rapid dilution with clean, dry air reduces the soot concentration, thereby limiting particle coagulation. By varying the flow rates of fuel, oxidizing air, and mixing nitrogen,

- it is possible to modify the characteristics of the flame, particularly its temperature, which affects both the quantity and the properties of the produced soot particles (Moore et al., 2014). As in the study by Marhaba et al. (Marhaba et al., 2019), propane flow is maintained at 60 mL/min, nitrogen flow at 0 mL/min, dilution air flow at 20 L/min and quenching nitrogen flow at 7 L/min. Variations of the soot production were obtained by varying the oxidation air flow rate from 0.9 L/min up to 1.5 L/min (0.9, 1, 1.15, 1.2, 1.25, 1.3, 1.35 and 1.5 L/min). The measurement points with oxidation air flow rate at 1, 1.15 and 1.5 L/min
- correspond respectively to the CAST3, CAST2 and CAST1 operating points in the study of Marhaba et al. (Marhaba et al., 2019). These specific operating points were also chosen for different studies (Bescond et al., 2016; Ouf et al., 2016). During the tests, stable generation points were sought, and before initiating filter sampling, the stability of the concentration production was ensured through measurements taken with online instruments such as the PPS and TEOM. Each measurement point was repeated three times, and the error bars represent the standard deviation of these measurements.
- The first heated line connecting the mini-CAST soot generator to the filter sampler is maintained at 180°C and is dedicated to sampling on quartz fiber filters (Whatman quartz filter, Grade QM-H, diam. 47 mm), for ex-situ gravimetric measurements, and for later OC/TC analysis. These filters are used for mass quantification of soot concentration during stable generation

conditions. Quartz filters are preconditioned by heating at 180°C in an oven for one hour to eliminate moisture. They are then weighed and stored with desiccant to prevent reabsorption of humidity. During measurements, the filters are placed in the

- 47 mm sampler with a 180°C temperature regulated heating jacket surrounding it. The collected mass is determined using a precision balance (Kern ABT 100-5M). The volume of gas passing through the filter is measured using a Gallus gas volumetric counter (see Fig 1.). Between the mass sampler and this Gallus volumetric counter, a heat exchanger cools the gas down to the ambient temperature, maintained at 20°C in the laboratory. By combining the collected mass with the measured gas volume, the soot mass concentration is calculated. This measurement is referred to as "Gravimetric mass concentration" in the study.
- The second method for determining the mass concentration of generated soot is based on the Sunset Lab OC/TC field instrument (Sunset Laboratory), which has been used for both the measurement of the OC/TC ratio and the mass concentration. In this instrument, to separate the organic fraction of carbon (OC) from the elemental fraction (EC), the samples are subjected to various temperature plateaus (up to 850°C) in a helium inert atmosphere for the OC fraction and in an oxidizing atmosphere for the EC fraction. The gases formed are then transported by the helium flow to a catalytic furnace, where they are oxidized
- to  $CO_2$ , then reduced to  $CH_4$  for more accurate measurement by a calibrated Flame-Ionization Detector (FID). A thermooptical correction is applied to separate OC from EC. The IMPROVE A protocol (Chow et al., 2001) has been used to reduce these pyrolytic conversions. OC/TC measurements have been performed on quartz fiber filters. Three punches of 1.5 cm<sup>2</sup> were analysed for each sample.

The second sampling line, after dilution, feeds real-time measurement instruments. The Dekati E-dilutor pro dilution system

- enables 2-stage dilution with a dilution factor set at 100. The first dilution stage is heated, while the second dilution stage operates at room temperature, where the aerosol sample is also cooled in a controlled manner. Both dilution stages are ejector-type, with a complementary sweep air flow in the duct. Concentrations downstream of the dilution system are compatible with the measurement ranges of the instruments used: a TEOM for a robust measurement of mass concentration, a PPS-Pegasor and a MA300 Aethalometer for real time mass concentration measurement and a SMPS for measurement of particle size
- distribution.

The TEOM (Thermo Scientific 1405) can continuously measure the mass of PM accumulating on a filter mounted upon an inertial microbalance using tapered element oscillating at its natural frequency (at 200 Hz). Aerosol is drawn in by a pump connected to the base of the microbalance. Sampled particles retained by the filter increase the mass of the oscillating system, producing a decrease in the natural frequency of vibration. Changes in the frequency of oscillation which are related to the

- mass of material accumulating on the filter are detected in quasi-real-time and converted by a microprocessor into an equivalent PM mass concentration (Allen et al., 1997). The sampled particles are heated at a temperature of 50°C. The PPS is a real time sensor that can be used to provide the mass and number concentration of aerosols in the exhaust of car engine based on electrical measurement. The device is supplied with clean, dry compressed air. In the first phase, the clean air is ionized by the corona effect and ejected through an orifice (Ntziachristos et al., 2013). Particles are electrically charged by
- the binding of ions within a zone isolated by a Faraday cage. Only the small number of ions that charged the particles surface are lost to the sensor outlet where an electrometer is used to measure the "compensation" electrical current. It is related to the

particle number concentration and particle size, and (Ntziachristos et al., 2004) demonstrated this instrument provides a measure of the active surface area of the studied aerosol (Rostedt et al., 2014).