# Peer review of "Mass concentration intercomparison of soot generated with Mini-Cast"

_Aerosol Research, 2025_

## Referee Comment (RC1)

**Review on manuscript ar-2025-15 "Mass concentration intercomparison of soot generated with Mini-Cast"**

This article compares several experimental methods for determining, directly or indirectly, the mass concentration of soot particles produced by a propane/air premixed flame. A mini-CAST generator was used and the total mass concentration of particles produced by this generator was determined by sampling and weighing on filters. The mass concentrations thus obtained for different operating points of the mini-CAST are then used to compare the mass concentrations determined by four methods with very different detection principles (thermo-optical analysis, analysis of the charge carried by the particles, optical analysis and extrapolation from an analysis of the particle size distribution obtained by electrical mobility analysis).

Although the experimental developments appear to have been carried out with great care and the results clearly presented, **a number of questions remain concerning the interest of such an article for the scientific community**, the generation of the mini-CAST and the comparison of various soot characterisation instruments that have already been the subject of numerous works:

- https://www.tandfonline.com/doi/epdf/10.1080/02786826.2010.482113?needAccess=true
- https://www.tandfonline.com/doi/epdf/10.1080/02786820701197078?needAccess=true
- https://doi.org/10.1089/ees.2014.0038
- https://doi.org/10.1021/es051228v

In addition, the assumptions associated with certain methods, and in particular the extrapolation from data obtained by the SMPS, are questionable, as is the transposition of the conclusions of this study to other sources of soot.

As it stands, this work, while of genuine technical quality, does not seem to me to be truly innovative, as it does not propose any new analytical strategies or corrections to be applied to the technologies investigated. What's more, the number of techniques is limited to the capacities of the two laboratories involved and does not allow us to cover a sufficient number of analysis technologies and instruments of the same technology in order to rule on possible sources of variability inherent in the different methods targeted.

**I do not recommend this manuscript as a research article for publication in the journal 'Aerosol Research'** and I invite the authors to submit this article in the form of a technical note.

Nevertheless, and in support of the quality of the technical work carried out and presented in this article, here are a number of comments that I feel are important to consider.

**Specific comments**

- **Abstract:** The authors mention that SMPS is an 'offline' method for determining mass concentration. As SMPS performs an on-line analysis of the particle size distribution, I do not think it is appropriate to mention this technique as an 'offline' method. SMPS softwares are also generally capable of directly converting particle size distributions by number into size distributions by mass (assuming spherical particles with a constant density, which is of course not relevant for soot particles), so the measurement is indeed "online".
- **2. Experimental setup:** SMPS specifications are missing, please add them;
- **Line 68-69:** the authors mention that the measurements were carried out 3 times and that the error bars in the graphs correspond to these repetitions, but the uncertainty inherent in the measurement process (in particular the measurement of mass concentration by weighing) is not evaluated, presented or discussed in the context of this comparison of methods;
- **Figure 1:** the impact of the transport line heated to 180°C, upstream of the filter sampler, on the determination of mass concentration by weighing and thermo-optical analysis was not discussed. One might wonder about a significant effect for samples with high OC/TC values. Have the thermograms been obtained and compared with and without this heated line to ensure that no volatile fraction is desorbed

under these conditions? This point is important as the sample is not heated for the line upstream of the dilution system;

- Still in connection with the impact of this line heated to 180°C, were SMPS size distributions or electron microscopy images taken before and after conditioning at 180°C? These questions are intended to shed light on the comparability of samples weighed on quartz filters and those characterised downstream of the dilution system;

- Has the actual dilution factor been evaluated for the different generation conditions? It is legitimate to wonder about possible particle losses within the dilution system and whether these losses differ according to the miniCAST settings. This point should be discussed and the uncertainty associated with determining this dilution factor should be taken into account when calculating the mass concentrations obtained downstream of the DEKATI diluter.

- **Line 77:** a heat exchanger is mentioned but not visible on figure 1, please add it;

- **Line 77:** what methodology (standard, standardised protocol) was used to determine the mass concentration from sampling on quartz filter? Has an assessment of the uncertainties (taking into account the uncertainties inherent in weighing and measuring the volume sampled) been carried out? If so, it should be added to the standard deviation associated with the three repetitions shown in Figures 4 to 8;

- **Figures 2 and 3** are not useful, as principles of the PPS and MA 300 instruments have been documented elsewhere and the present article does not introduce any significant changes to the principle of these instruments;

- **Lines 114 to 124** seem to me to go into too much detail and could concentrate on the fact that the PPS mass calibration constant was established essentially on automobile emissions with a relatively limited range of size distribution in numbers;

- **Figure 4 right:** the characterisation of the mini-CAST size distribution has already been the subject of numerous publications, so it does not seem useful to illustrate this with a figure (or at least to place this figure in an appendix to the article);

- With regard to **bimodal distributions**, are TEM images of the particles available in order to determine whether they are bimodal?

- **Table 1:**
    o The first line requires an explanation of the 1% mentioned after the mode at 24.5 nm. If this is a monomodal distribution, shouldn't it be 100%?
    o On what criteria do the authors conclude that the particle size distributions produced within the range of oxidation flow rate 1 to 1.35 L.min-1 are bimodal? Is monomodal smoothing totally unsuitable and on what criteria was bimodal smoothing preferred?

- **Figure 5:** have the measurement uncertainties of the total carbon mass concentration based on thermo-optical analysis been determined?

- **Line 196 :** "we considered a so-called true density for the aggregates that varies depending on the considered point" ➔ clarification of the method used to determine the mass concentration from the particle size determined by SMPS is required. Did the authors consider a single density value for the entire particle size distribution? If so, a discussion appears necessary and must be confronted with the notion of effective density of soot particles. Conversion models, including the fractal morphology of soot, have been developed and are available in the literature. Why not consider them in this article to achieve a conversion from number to mass size distributions?
    o https://doi.org/10.1016/j.carbon.2024.119197
    o https://www.tandfonline.com/doi/full/10.1080/02786826.2019.1577949
    o https://www.sciencedirect.com/science/article/abs/pii/S0010218018304310
    o https://www.sciencedirect.com/science/article/abs/pii/S0021850215000701
    o https://www.sciencedirect.com/science/article/pii/S0021850223000769#sec5

- **Line 211:** the following sentence: "We report in table 2 the values obtained for the OC/TC ratios as determined by thermo-optical analysis and the corresponding evaluated true densities, that were used to evaluate the SMPS mass concentrations reported on Fig. 7" is not clear;

- **Lines 213-221:** this paragraph questions the capacity of the mini-CAST to be act as a reference generator, the authors stating at the beginning of their article that they wanted to use this generator for its stability.

If the composition of soot varies from one mini-CAST to another, how can the conclusions of this article be generalised to other mini-CASTs

- **Table 2:** It is not easy to know whether the density values given in this table are calculated or determined to give the best agreement with the weighing measurement;
- **Lines 224-228**: this sentence is not easy to understand;
- **Figure 8, left**: error on the x-axis legend « Gravimteric »
- **Figure 8, right**: I don't think this figure is useful, as the OC/TC ratio can be mentioned in the left part of the figure.
- **Figure 8 right**: oxydation ➔ oxidation

---

## Author Comment (AC1)

**Review on manuscript ar-2025-15 "Mass concentration intercomparison of soot generated with Mini-Cast"**

This article compares several experimental methods for determining, directly or indirectly, the mass concentration of soot particles produced by a propane/air premixed flame. A mini-CAST generator was used and the total mass concentration of particles produced by this generator was determined by sampling and weighing on filters. The mass concentrations thus obtained for different operating points of the mini-CAST are then used to compare the mass concentrations determined by four methods with very different detection principles (thermo-optical analysis, analysis of the charge carried by the particles, optical analysis and extrapolation from an analysis of the particle size distribution obtained by electrical mobility analysis).

Although the experimental developments appear to have been carried out with great care and the results clearly presented, **a number of questions remain concerning the interest of such an article for the scientific community**, the generation of the mini-CAST and the comparison of various soot characterisation instruments that have already been the subject of numerous works:
- https://www.tandfonline.com/doi/epdf/10.1080/02786826.2010.482113?needAccess=true
- https://www.tandfonline.com/doi/epdf/10.1080/02786820701197078?needAccess=true
- https://doi.org/10.1089/ees.2014.0038
- https://doi.org/10.1021/es051228v

In addition, the assumptions associated with certain methods, and in particular the extrapolation from data obtained by the SMPS, are questionable, as is the transposition of the conclusions of this study to other sources of soot.

As it stands, this work, while of genuine technical quality, does not seem to me to be truly innovative, as it does not propose any new analytical strategies or corrections to be applied to the technologies investigated. What's more, the number of techniques is limited to the capacities of the two laboratories involved and does not allow us to cover a sufficient number of analysis technologies and instruments of the same technology in order to rule on possible sources of variability inherent in the different methods targeted.

*Thank you for your deep and pertinent review. In the following answers, we have tried to enhance the clarity and impact of the demonstration of the scientific relevance of the paper. We have modified the title of the paper and brought some improvements. The new proposed title is "Evaluation of mass measurement techniques for soot with different size distributions and OC/TC contents"*

**I do not recommend this manuscript as a research article for publication in the journal 'Aerosol Research'** and I invite the authors to submit this article in the form of a technical note.

Nevertheless, and in support of the quality of the technical work carried out and presented in this article, here are a number of comments that I feel are important to consider.

**Specific comments**
- **Abstract:** The authors mention that SMPS is an 'offline' method for determining mass concentration. As SMPS performs an on-line analysis of the particle size distribution, I do not think it is appropriate to mention this technique as an 'offline' method. SMPS softwares are also generally capable of directly converting particle size distributions by number into size distributions by mass (assuming spherical particles with a constant density, which is of course not relevant for soot particles), so the measurement is indeed "online".
*It has been modified in the abstract.*

- **2. Experimental setup:** SMPS specifications are missing, please add them; *It has been added in the text.*

- **Line 68-69:** the authors mention that the measurements were carried out 3 times and that the error bars in the graphs correspond to these repetitions, but the uncertainty inherent in the measurement process (in particular the measurement of mass concentration by weighing) is not evaluated, presented or discussed in the context of this comparison of methods;
*A paragraph has been added to the text to evaluate the gravimetric weighing based on the Iso 15767 standard.*

- **Figure 1:** the impact of the transport line heated to 180°C, upstream of the filter sampler, on the determination of mass concentration by weighing and thermo-optical analysis was not discussed. One might wonder about a significant effect for samples with high OC/TC values. Have the thermograms been obtained and compared with and without this heated line to ensure that no volatile fraction is desorbed under these conditions? This point is important as the sample is not heated for the line upstream of the dilution system;

*Indeed, this is an important point, and the aim of this experiment was to test the filter sampling procedure that is applied in direct exhaust measurements, particularly for ship engines. In such exhaust measurements where temperatures are over 200 °C depending on the sampling location, the main issue is to avoid condensation of*

moisture and semi-volatile species. The procedure follows the ISO 8178 standard, which specifies a target temperature of 180 °C for the sampling line and a maximum residence time of 3s. With a sampling flow rate of 7.7 L/min, a line length of 1 m, and an inner diameter of 6 mm, the aerosol residence time is approximately 0.22 s. This duration is extremely short compared to the 173 s of the first temperature plateau at 140 °C on the thermogram shown below, which is required to vaporize only 1.3% of the total carbon mass.

In the revised presentation of the results from this study, the TEOM measurement is considered the reference. All measurements are performed downstream of the dilution system (TEOM, SMPS, PPS, MA300), except for the gravimetric measurement, whose results are validated by comparison with the TEOM measurements, as discussed later in the article.

**-** Still in connection with the impact of this line heated to 180°C, were SMPS size distributions or electron microscopy images taken before and after conditioning at 180°C? These questions are intended to shed light on the comparability of samples weighed on quartz filters and those characterised downstream of the dilution system; All instruments are positioned behind the dilution system, so at iso sampling conditions, except the filter mass sampling. We provided elements on the previous question/remark, and believe they clarify the comparability.

**-** Has the actual dilution factor been evaluated for the different generation conditions? It is legitimate to wonder about possible particle losses within the dilution system and whether these losses differ according to the miniCAST settings. This point should be discussed and the uncertainty associated with determining this dilution factor should be taken into account when calculating the mass concentrations obtained downstream of the DEKATI diluter. All the instruments used in this study are positioned downstream of the dilution system and are therefore all potentially affected by biases introduced by the dilution process, with the exception of the gravimetric filter-based sampling, as previously mentioned.

**- Line 77:** a heat exchanger is mentioned but not visible on figure 1, please add it; Thank you. It has been added to the article.

**- Line 77:** what methodology (standard, standardised protocol) was used to determine the mass concentration from sampling on quartz filter? Has an assessment of the uncertainties (taking into account the uncertainties inherent in weighing and measuring the volume sampled) been carried out? If so, it should be added to the standard deviation associated with the three repetitions shown in Figures 4 to 8; Indeed, the standard procedure implemented is already described in the manuscript, but we now provide a more detailed explanation of the uncertainty calculation, which is based on the combined uncertainties associated with weighing and sampled volume measurement. For this uncertainty evaluation, we followed ISO 15767 to assess the uncertainty related to weighing, evaluated the uncertainty on the sampled volume, and applied uncertainty propagation to estimate the overall uncertainty, which combines measurement uncertainty with the uncertainty associated with repeatability based on the three repetitions. The error bars shown in Figure 5 for the "Gravimetric mass conc." result from this analysis, while the uncertainty in the determination of "Total Carbon (TC) mass conc." determined through the thermo-optical analysis is estimated at 13%, as established through the literature review.

[Figure]

**- Figures 2 and 3** are not useful, as principles of the PPS and MA 300 instruments have been documented elsewhere and the present article does not introduce any significant changes to the principle of these instruments. We wanted the reader to find all the operating principles of the devices used in this article without having to resort to additional bibliography.

**- Lines 114 to 124** seem to me to go into too much detail and could concentrate on the fact that the PPS mass calibration constant was established essentially on automobile emissions with a relatively limited range of size distribution in numbers; It has been modified in the text.

**- Figure 4 right:** the characterisation of the mini-CAST size distribution has already been the subject of numerous publications, so it does not seem useful to illustrate this with a figure (or at least to place this figure in an appendix to the article);
In the bibliography, the intermediate points have not been investigated. It is interesting to show the intermediate size distributions.

**- With regard to bimodal distributions**, are TEM images of the particles available in order to determine whether they are bimodal?
No TEM images have been done. The study was conducted at eight different oxidation air flow settings and involved several instruments or analytical protocols requiring significant analysis time. The analysis and statistical processing of the large number of TEM images that would have been required did not seem reasonable to us, although such an approach would indeed have enriched the experimental investigation.

**- Table 1:**
o  The first line requires an explanation of the 1% mentioned after the mode at 24.5 nm. If this is a monomodal distribution, shouldn't it be 100%?
Thank you. It has been corrected in the article

o  On what criteria do the authors conclude that the particle size distributions produced within the range of oxidation flow rate 1 to 1.35 L.min-1 are bimodal? Is monomodal smoothing totally unsuitable and on what criteria was bimodal smoothing preferred?

The procedure implemented to perform a least-squares minimization aimed at representing the measured distribution as the sum of two log-normal distributions is presented on the left-hand side. On the right, it is clearly visible that a single log-normal distribution fails to accurately represent the measured distribution produced by the miniCAST at the 1.25 L/min oxidation air flow condition.

[Figure]

**- Figure 5:** have the measurement uncertainties of the total carbon mass concentration based on thermooptical analysis been determined?
For the Total Carbon (TC) mass concentration based on thermo-optical analysis, Sipkens et al. (2024) evaluated the TC uncertainties to 13%, based on bibliography. Brown et al. (2017) reported combined standard error below 13% for a reproducibility between four laboratories and Schmid et al. (2001) reported an uncertainty on their TC measurement between 6.7% and 11%. We therefore decided to take the conservative value of 13%, as reported in Figure 5.

**- Line 196 :** "we considered a so-called true density for the aggregates that varies depending on the considered point" ➜ clarification of the method used to determine the mass concentration from the particle size determined by SMPS is required. Did the authors consider a single density value for the entire particle size distribution? If so, a discussion appears necessary and must be confronted with the notion of effective density of soot particles. Conversion models, including the fractal morphology of soot, have been developed and are available in the literature. Why not consider them in this article to achieve a conversion from number to mass size distributions?
o  https://doi.org/10.1016/j.carbon.2024.119197
o  https://www.tandfonline.com/doi/full/10.1080/02786826.2019.1577949
o  https://www.sciencedirect.com/science/article/abs/pii/S0010218018304310
o  https://www.sciencedirect.com/science/article/abs/pii/S0021850215000701

o https://www.sciencedirect.com/science/article/pii/S0021850223000769#sec5

Indeed, other approaches have been developed to estimate an "effective density" of soot particles, considering their fractal nature. And based on an analysis of various data sets available in the literature, Olfert and Rogak proposed a model that expresses the effective density as a function of mobility diameter. This model allows the evaluation of the decrease in effective density with the mobility diameter of soot aggregates and is expressed using two parameters: a negative exponent and a reference density defined at a mobility diameter of 100 nanometers. The data used to develop this model originates primarily from studies of particles produced by internal combustion engines or gas turbines, and a value of 510 kg.m$^{-3}$ for the effective density at 100 nm was proposed for this model. However, Yon et al. reported higher effective density values at 100 nm for the miniCAST and reported values ranging from 1100 to 765 kg.m$^{-3}$ for oxidation air flow rates between 1 and 1.5 L/min. Although only three operating points were investigated, the decrease in effective density with increasing oxidation air flow rate was found to be linear. And we observed that the decay in OC/TC with oxidation air flowrate is also linear. We therefore conducted a 'best fit' identification procedure to obtain the values of $\rho_{eff,100}$, the density at 100 nm that would allow us to recover the TEOM reference mass concentrations from the SMPS analysis together with the Olfert and Rogak model. We obtained a set of values $\rho_{eff,100}$ "Best fit analysis" (see Fig. 7 right) and observed that the change in this density at 100 nm with the OC/TC ratio that can be represented by a linear interpolation, expressed as:

$$\rho_{eff,100}(x) = 932 + 661 \cdot x \tag{4}$$

with x the OC/TC ratio.

[Figure]

**Figure 7 – Evolution of the OC/TC ratio as a function of the oxidation air flowrate (left). Identification of $\rho_{eff,100}$ as a function of OC/TC ratio.**

From this Equ. (4), interpolated values of $\rho_{eff,100}$ can be evaluated. These effective densities at 100 nanometers determined through this procedure are reported in Table 2, and the corresponding mass concentrations, labeled "SMPS effective density," are shown in Fig. 8: the use of this effective density model combined with the measured size distributions results in a good estimation of the mass concentration.

[Figure]

**Figure 8 – Comparison of gravimetric mass concentrations with evaluations based on PPS and SMPS measurements.**

**- Line 211:** the following sentence: "We report in table 2 the values obtained for the OC/TC ratios as determined by thermo-optical analysis and the corresponding evaluated true densities, that were used to evaluate the SMPS mass concentrations reported on Fig. 7" is not clear;

Indeed we clarified this sentence. But first we mentioned earlier in the text that the density is considered as constant for a given point of the mini cast: "For this evaluation, we initially considered a constant so-called "true density", independent of the particle mobility diameter, but varying with the OC/TC ratio and therefore dependent on the selected operating point of the miniCAST"

Then the sentence itself is clearer and we explain that the true density is calculated.

"We report in table 2 the values obtained for the OC/TC ratios as determined by thermo-optical analysis and the corresponding evaluated true densities calculated with the Park mixing model (Eq. 2), that were used to evaluate the "SMPS true density" mass concentrations reported on Fig. 7"

**- Lines 213-221:** this paragraph questions the capacity of the mini-CAST to be act as a reference generator, the authors stating at the beginning of their article that they wanted to use this generator for its stability. If the composition of soot varies from one mini-CAST to another, how can the conclusions of this article be generalised to other mini-CASTs
Indeed, the "S" in miniCAST was introduced by its designer to indicate "standard." It is therefore a generator that is intended to be reproducible. However, the setup installed downstream of the generator can lead to variations in the soot size distributions, due to mechanisms such as agglomeration and deposition. Moreover, to our knowledge, no intercomparison between different miniCAST generators is available, so strict equivalence from one generator to another cannot be guaranteed. Nevertheless, we emphasize here the generator's stability, which is well established: it can produce soot with stable concentration and size distribution over relatively long periods, thus enabling gravimetric measurements and downstream characterization under dilution, using a steady source.

**- Table 2:** It is not easy to know whether the density values given in this table are calculated or determined to give the best agreement with the weighing measurement.
Indeed, we now detail that "true densities calculated with the Park mixing model (Eq. 2)". And it is also mentioned in table 2 itself. Since an effective density is also used now, it is also mentioned that it is calculated with Equ. 4.

**- Lines 224-228:** this sentence is not easy to understand;
Indeed, we tried to clarify, and modified the sentence, as proposed below

The measured values of the OC/TC organic fraction contained in soot compare well with those obtained in the previous study by (Marhaba et al., 2019a) for equivalent operating conditions of the Mini-CAST. For the considered points named CAST1, CAST2 and CAST3 in Marhaba et al. study (CAST3 corresponds to 1 L/min, CAST2 to 1.15 L/min and CAST1 to 1.5 L/min of oxidation airflow), Marhaba et al. reported OC/TC ratio values of 87%, 46.8% and 4.1%, respectively. In the same operating conditions, we measured corresponding OC/TC values of respectively 56.7%, 46 % and 6.2%. For the point CAST3, our measured value is significantly lower, while the two other points are coherent. However, as highlighted by (Moore et al., 2014), soot production conditions can vary with parameters other than the overall carbon/oxygen ratio of the flame, and variations in mode size or OC/TC ratio have already been observed between different studies using different Cast or Mini-CAST generators.

**- Figure 8, left:** error on the x-axis legend « Gravimteric »

Thank you.  It has been modified.

**- Figure 8, right:** I don't think this figure is useful, as the OC/TC ratio can be mentioned in the left part of the figure.
We understand your recommendation to limit the number of figures. However, to maintain the clarity, we were not able to modify the manuscript in this direction.

**- Figure 8 right:** oxydation ➔ oxidation :

 Thank you. It has been corrected in the article.

---

## Author Comment (AC2)

**Review R2**

The manuscript describes an intercomparison of several instruments for measuring particle mass from the output of a Mini-CAST generator. The comparison includes both offline and online instruments based on various measurement principles. The results from the instruments are compared against offline gravimetric determination of the particle mass from a filter deposition. An offline OC/TC analysis is also performed to support the analysis of the results. The Mini-CAST is run with varying amounts of oxidation air flow to achieve different levels of organic content, total mass, and different particle size distributions. A potential complication here is that all the parameters are changed together, which can complicate the analysis, but the authors make convincing arguments for the dependence of the measurement results on the chancing parameters. The different methods show good agreement, to approximately 10% from the reference values, with the caveats that PPS requires a size dependent correction and with MA300, different wavelengths correlate with the total particle mass, depending on the OC/TC ratio.

I believe the topic is suitable for the journal and would be of interest to its readers. The measurements and the analysis are described thoroughly, and support the conclusions mostly, but there are a few unclear points I would ask the authors to address. I have also listed a few stylistic or typographical points at the end.

line 182: Any comments on why the PPS correctly estimates the largest particles from Mini-CAST, if the size distribution from the engines, which were used in the calibration, is mostly smaller particles (according to the geometric mean at line 119)?

The PPS measures leakage current based on the electrical charge of an entire particle size distribution. Its calibration was performed using aerosols generated by a diesel engine, which were variable in concentration but stable in terms of median diameter—monomodal distributions centered at 48 nm with a geometric standard deviation of 1.78. In our study, the aerosol characteristics are more variable, including parameters such as the OC/TC ratio and the shape of the size distribution, which ranges from monomodal to strongly bimodal. We observe that as the median diameter of the distribution decreases, the OC/TC ratio increases significantly. For instance, even with a median diameter close to 48 nm, OC/TC ratios can already reach high levels, around 40%. However, no information is available regarding the OC/TC ratio under the PPS calibration conditions. A high organic fraction may alter the morphology of the aggregates, and consequently the surface electrical charging, which is the basis of the PPS measurement principle.

We changed this sentence to "the PPS mass calibration constant was established essentially on automobile emissions with a relatively limited range of size distribution in numbers"

We also added uncertainty on this figure to consider all experimental uncertainties.

figure 6 (right): I don't quite understand the bottom axis here. Should it not match the last column of table 1? For example, the point marked 1.2 has a median diameter of roughly 60 nm in table 1, but in the figure it looks to be less than 50 nm. The point marked 1.25 also appears to be out of place in the figure. It's "median diameter" in the figure is closer to the mode 1 diameter from table 1, to my eye. The axis title is also unclear.

Indeed, there was an update that was not made in the version submitted, and we took that into account. Thank you, it has been corrected in the article project.

line 189: The statement that the concentration ratio of PPS/gravimetric is independent of the median diameter may be a little misleading, since the diameter is very close to each other in this range of oxidant flow rate. That is, based on the table 1 data, unless the figure 6 data is correct and I have misunderstood something here.

Indeed, we discussed this point in your first comment. We think we have answered, and we also believe it must be limited to the air range mentioned. The sentence has been updated to:

 "The raw PPS measurement becomes reliable for soot generated with an oxidation air flowrate in the limited range of 1.3 to 1.5 L/min, where the mass concentration ratio between the PPS and the TEOM reference method remains nearly constant and close to 1, regardless of the measured median diameter."

line 238: Did you try to evaluate the MA300 data from the different wavelength channels together? For example, could you try to estimate the total mass from the BC and BrC fractions together, with less dependence on the OC/TC ratio?

Indeed, the purpose of using multiple wavelengths is to access more detailed information about the composition of carbonaceous particles. It is known that the 880-nanometer wavelength corresponds to a maximum response from the so-called black carbon fraction, and it is also known that the presence of BrC—representing a significant organic fraction—can be detected at the 375-nanometer wavelength. We could not exploit the ultraviolet results to account for BrC. However, by correcting the results obtained in the infrared—thus corresponding a priori only to the black carbon portion—using the formula TC = EC × (1 + OC/EC), we obtain a diagram like the one shown below. The proposed correction, through the consideration of the OC/EC ratio, indeed allows for a correction of the mass concentration referred to as black carbon. But the limit here is that the correction is based on an thermos-optical measurement technique.

[Figure]

Minor comments:

line 45: Although it becomes clear from context later, it is unclear here, what instrument the sentence "and considered as the reference measurement" refers to.

Indeed, in the revised version and based on the comments of R1 and R3, we have evaluated the mass concentration downstream the diluter with different instruments and finally considered the TEOM as the reference instrument. The manuscript is revised as:

"The mass concentration evaluated by the online devices (MA300, PPS Pegasor, and SMPS) and the offline direct gravimetric measurement were studied as a function of the reference mass concentration evaluated by the TEOM. The evaluation of the OC/TC ratio done from the filters, and the size distributions of the produced soot particles are also necessary for the interpretation of the instrument responses."

line 47: you should mention the OC/TC measurement done from the filters here. Right now, it sounds like SMPS could do that as well.

Indeed, the sentence has been clarified.

figure 5: I would suggest changing the legend entry for "Improve A" to "OC/TC analysis" or mentioning the protocol also in the figure caption.

Indeed, the figure caption has been modified.

figure 7 (left): there seems to be a typo in the bottom axis title.

Thank you, it has been corrected.

---

## Author Comment (AC3)

**Review R3**

The article compares the mass concentration of the mini-CAST soot generator based on an experiment that evaluates both online and offline measurement methods. The primary motivation for the study appears to be the determination of correction calibration parameters for mass calibration of the PPS instrument, which is based on aerosol active surface area.

Although the experiments were conducted with technical accuracy, I believe the experimental design should have been more robust. In addition, the conclusions drawn are **overly general and simplistic**. Therefore, **I do not recommend that this manuscript be accepted for publication** in Atmospheric Research.

Below, I outline several points that do not meet the standards of a research article and should instead be considered for publication as a technical report in a more appropriate journal once the following revisions are implemented.

1. Experimental setup, heated sampling line, and mass of PM from OC/EC analysis:

The experimental setup, as described in the text, states: "This setup includes the soot generation source, the mini-CAST, followed by two heated lines maintained at 180 °C. One line is used for filter sampling, while the other feeds a dilution system that distributes the diluted and cooled aerosol to various measurement instruments: the TEOM, PPS, MA300, and SMPS." However, the schematic only shows a heated line leading toward the offline sampling system. This raises an immediate questions: **Was the heated line actually used in both branches (online and offline), or only in the offline branch? Why was a heated line used at all, given that it can significantly alter the chemical and physical properties of aerosols**?

Indeed, this is an important point, and the aim of this experiment was to test the filter sampling procedure that is applied in direct exhaust measurements, particularly for ship engines, where there is distance between the sampling probe and the collection filter holder. We need a flexible connection (as short as possible, and with a length of here of 1m) to open the holder and change filters. It must be heated to avoid condensation. In such exhaust measurements where temperatures are over 200 °C depending on the sampling location, the main issue is to avoid condensation of moisture and semi-volatile species. The procedure follows ISO 8178, which specifies a target temperature of 180 °C for the sampling line and a maximum residence time of 3 seconds.

For comparison, if the OC/EC analysis followed the IMPROVE protocol—where the OC1 fraction is defined at 140 °C—this fraction would be entirely lost within a 180 °C heated line. Furthermore, heating the sampled air can lead to misinterpretation of pyrolyzed carbon (PC) in thermal-optical OC/EC analysis, as the initial reflectance from the filter may already include pyrolyzed material, resulting in an overestimation of elemental carbon (EC).

If one sampling branch was heated and the other was not, this introduces a fundamental inconsistency that compromises the comparability of the results.

With a sampling flow rate of 7.7 L/min, a line length of 1 m, and an inner diameter of 6 mm, the aerosol residence time is approximately 0.22 s. This duration is extremely short compared to the 173 s of the first temperature plateau at 140 °C on the thermogram shown below, which is required to vaporize only 1.3% of the total carbon mass.
In the revised presentation of the results from this study, the TEOM measurement is considered the reference. All measurements are performed downstream of the dilution system (TEOM, SMPS, PPS, MA300), except for the gravimetric measurement, whose results are validated by comparison with the TEOM measurements, as discussed later in the article.

Additionally, the authors should provide a more detailed explanation of how they calculated mass concentration based on OC/EC analysis. Assuming that the total aerosol mass (PM) is equivalent to the mass of carbonaceous aerosols (CA), the following equation applies:

PM=CA= OA+EC= TC·(OA/OC)−EC·(OA/OC−1)

where:

OC = OC1 + OC2 + OC3 + OC4 + PC

EC = EC1 + EC2 + EC3 – PC

OA/OC is the organic aerosol to organic carbon ratio, which can be obtained either through complementary measurements (e.g., ACSM for OA, OC/EC analysis for OC) or sourced from literature.

This work is carried out within a context of mass concentration measurement in field measurement campaign, evaluated in laboratory conditions. The ACSM is designated for long-term monitoring to study the temporal changes and variability of OA sources. In our case, we used a mini-cast generator to generate carbonaceous soot particles, and the chemical composition was not a point we tried to investigate, knowing that an ACSM could be difficult to implement on future field campaigns. We sampled soot on a filter and analyzed it with the Sunset Laboratory Analyzer. We did not make any real-time measurements.

What OA/OC ratio was used in the mass estimation?

We estimated only the OC/TC using the Sunset Semi-Continuous Organic Carbon/Elemental Carbon (Sunset OC/EC) aerosol analyzer that utilizes the modified National Institute for Occupational Safety and Health thermo-optical method to determine total carbon (TC), organic carbon (OC), and elemental carbon (EC) at near real-time.

Was this ratio measured directly or adopted from previous studies? How do the authors justify the use of a heated line, and how do they address the uncertainties associated with the loss of the OC1 fraction and the potential misclassification of pyrolyzed carbon in thermal-optical analysis?

As mentioned above, the residence time in this short, heated line is no more than 0.22s in the condition of the experiment. In addition, we observed that the volatilization time at this temperature of 180°C is 184s, for a total vaporization of 0.8 µg/cm$^2$ over a total TC of 57.53 µg/cm$^2$ that represent 1.3% of the total mass, and 2% of the organic fraction OC.

With a residence time of 0.22s, the corresponding vaporized fraction would be around 0.002 %.

[Figure]

2. Experimental Design and Showing  Method Equivalence

In demonstrating the suitability of using the PPS, two critical steps are missing. The reader is concerned by the simultaneous variation of **two parameters at the aerosol source—namely, the mass concentration and the aerosol composition as a function of the oxidation flow rate in the miniCAST**. This affects the OC/TC ratio

and introduces uncertainty. Additionally, there is a lack of evidence demonstrating the equivalence between the **reference gravimetric method and the TEOM online method**.

As a result, the reader cannot apply the calibration factors for the PPS in their own experiment, since two parameters are being altered simultaneously. Therefore, the experiment should be redesigned so that, for a fixed OC/TC ratio at the source, only the mass concentration is varied. This would allow for the determination of calibration factors under controlled conditions. This procedure should be repeated for multiple OC/TC ratios. Only in this way can the reader reliably apply the calibration factors in their experimental setup.

Thank you for your suggestion. In this paper we varied the oxidation air flow rate, which is the main operating parameters of the miniCAST that is found to be changed in most available studies using this generator. Changing this parameter indeed leads to the variation of two parameters at the same time, namely the OC/TC ratio and the mass concentration. As suggested, using a pure dilution of the generated soot on single miniCAST generation points would have been relevant to study the mass concentration linearity response of instrument. In another hand, changing the OC/TC ratio was interesting to study the aethalometer MA300 response. This discussion is interesting for future study. We added this suggestion to the work perspectives and as a discussion point on the present study, in conclusion:

"The results of this study show that the selected operating points led to simultaneous variations in mass concentration, size distribution, and OC/EC ratio. It would be relevant, by varying other gas flow rates of the miniCAST or the aerosol dilution conditions, to better decouple the parameters under investigation in future studies."

Furthermore, prior to this, the equivalence between the TEOM and the offline gravimetric method for measuring mass concentration should be demonstrated. Establishing this equivalence would validate the TEOM as a reference method against which all other measurements (SMPS, MA300, PPS) can be compared. It would also confirm the correct operation of the diluter and eliminate concerns regarding potential heating effects in the sampling line. For the comparison between the offline gravimetric method and the TEOM, authors should apply, for example, equivalence assessment tools as outlined in the European standard EN 16450:2017, which specifies requirements for automated measuring systems for particulate matter, including slope and offset of the orthogonal regression assessment. Only after establishing this equivalence can the results from SMPS, MA300, and PPS be interpreted with confidence and used to draw scientifically robust conclusions.

Thank you for this detailed and precise review of our submission. We provide responses to the various questions and comments below: we have considered your comment and based on ISO 16450 we have evaluated the slope and offset of the orthogonal assessment. The linear regression established between the reference and tested measurements satisfies the ISO 16450 validation criteria, with the slope $b$ statistically compatible with unity ($|b - 1| < 2u_b$), and the intercept $a$ close to zero and statistically compatible with it ($|a| < 2u_a$), where $u_b$ and $u_a$ are the standard uncertainties associated with the slope and intercept, respectively. So, we have established the equivalence between the TEOM and the gravimetric measurement. Finally, we concluded that the TEOM should be considered as the reference measurement, being online in the exact same conditions downstream the dilution system.

[Figure]

3. Lack of critical evaluation in the interpretation of filter photometer measurement results

The comparison of mass concentration measurements based on attenuation using the MA300 filter photometer is overly superficial. When using filter photometers with laboratory-generated aerosols, it is essential to carefully consider filter loading corrections, and the C parameter (multiple scattering correction factor), which depends not only on the properties of the filter substrate but also on the optical characteristics of the aerosol.

Indeed, these parameters were not sufficiently detailed in the originally submitted version, where we moved rather directly to the results. We propose to enrich the revised submission by specifying several factors, including some of those you requested. As suggested, we will also expand the discussion regarding filter loading and the associated correction factors, which we may have addressed too briefly.

In the newly submitted version of the paper:

"The MA300 aethalometer (AethLabs, San Francisco) is a real time portable analyzer that uses the filter-based light absorption principle: the mass concentration measurement is based on the optical attenuation ATN of a light beam passing through a cake of particles collected on a filtering fiber tape. The instrument has five analytical channels that operate at different wavelengths (375, 470, 528, 625 and 880 nm), and the measurement of absorption at 880 nm is interpreted as Black Carbon (Hansen et al., 1984). The MA300 draws a controlled flow through the collection spot on the filter tape, and when the light attenuation reaches a threshold value, the tape advances automatically.

The attenuation ATN is defined (Gundel et al., 1984) as ATN = $-100 * \ln(I/I_0)$, with I and I0 respectively the measured intensity trough the loaded spot signal and the reference signal, through the empty filter (see Fig. 3). The instrument measures the transmission of light through the particle-loaded filter, from which the attenuation (ATN) is derived based on its rate of change over time. This attenuation is then converted into an absorption coefficient. Finally, the equivalent black carbon mass concentration is obtained by dividing the absorption coefficient by the mass absorption cross-section specific to black carbon (Drinovec et al., 2015). With this definition and for the 880 nm wavelength, the deposited Black Carbon BC mass concentration is then evaluated as:

$$BC = \frac{S \cdot (\Delta ATN / 100)}{F(1-\xi) \cdot \sigma \cdot C \cdot (1-k \cdot ATN) \cdot \Delta t} \qquad (1)$$

Where S is the collection spot area (for the MA300, S=0.0707 cm2) , DATN the changes of attenuation during interval Dt, F the measured flow rate, and Dt the time interval between two measurements (Dt was 5 sec. in this study).

Since the airflow is measured downstream of the filter, it is necessary to account for lateral air leakage within the filter matrix beneath the optical chamber, represented by the parameter $\zeta$, the leakage factor. The value of $\zeta$ is determined by comparing the inlet and outlet flows and is calibrated by the aethalometer manufacturer. Drinovec et al. reported values of $\zeta$ ranging from 2% to 7% (Drinovec et al., 2015). In Equ. (1), the $\sigma$ (m2/g) and C are the mass absorption cross-section and multiple scattering coefficient factor (Weingartner et al., 2003). Filter-based light absorption techniques are subject to measurement artifacts due to scattering on the filter (Weingartner et al., 2003). These empirical parameters are related to the instrument design and filter material (Wu et al., 2024). The C parameter considers multiple scattering correction factor, which mainly depends on the properties of the filter substrate, and on the optical characteristics of the aerosol deposited on the filter. According to Drinovec et al. (2015), C strongly depends on the filter material used. Indeed, the optical absorption of aerosols collected on a filter is affected by light scattering within the filter matrix, which is dependent on the filter material, but with no significant spectral dependence (Drinovec et al., 2015). The value of C is determined by comparing the results of different measurement methods in laboratories and in ambient observatories, and Drinovec et al. report values in a range of 1-1.36 (for tetrafluoroethylene TFE/Quartz with variable ratio) up to 2.14 (for Quartz filter). For our measurement instrument, the sampling filter tape is composed of PTFE (polytetrafluoroethylene), and the manufacturer calibrated value of C is 1.3. The MAC (Mass Absorption Cross-section) or Mass Absorption Coefficient is defined as $\sigma_{air} = \sigma_{ATN} / C$, with $\sigma_{ATN}$ the Specific Attenuation Cross-section, dependent on the wavelengths $\lambda$ ($\sigma_{ATN}$ =10.120 m2/g for IR at l=880 nm, up to $\sigma_{ATN}$ =24.069 for UV at l=375 nm).

One of the identified drawbacks of aethalometers is the overestimation of BC concentration on a new filter and the underestimation when the loading is high, with the most accurate concentrations being obtained on slightly loaded filters (Arnott et al., 2005; Collaud Coen et al., 2010). Several models have been proposed to consider this loading effect when processing aethalometers data, such as Weingartner et al. (2003) and Virkkula et al.

(2007) models, which have been most frequently used for loading effect compensation. Drinovec et al. (2015) developed a real-time method to compensate nonlinearity with high time resolution, by using a double collection on two spots in parallel at different flow rates. They introduced the new reference Aethalometer model AE33 (Drinovec et al., 2015). In this model, the k constant is real-time evaluated from the algorithm described in (Drinovec et al., 2015), for each wavelength, and this so-called "dual-spot technology" permits to eliminate the data artifact due to filter loading. AE33 is now considered one of the most popular instruments for the real-time measurements of aerosol BC. The aethalometer MA300 we used in this study is a portable version also using this patented "dual spot" technology that makes it possible to eliminate data distortions due to filter loading.

Aethalometers have been used to monitor black carbon concentrations in urban areas (Blanco-Donado et al., 2022), assess biomass burning contribution to emissions in urban areas (Favez et al., 2007), or to measure personal exposure in working environment (Gren et al., 2022). However, the ability of aethalometers to accurately quantify soot emissions remains an active research topic. The recent study by Aakko-Saksa et al. (Aakko-Saksa et al., 2022a) over a wide ship exhaust test matrix obtained with different fuels, engines, and emission control devices showed that measurement with an aethalometer could lead to overestimation of BC emissions."

These optical properties include aerosol coating effects, which may be transparent or light-absorbing, and cannot be adequately described by a simple OC-to-TC ratio, as suggested in the manuscript.

We are indeed aware of the effect described, and our intention is not to reduce the interpretation of results from this type of instrument to a single parameter. In the specific case of soot particles generated by propane combustion in the miniCAST burner, it is well established in the literature that the primary particle diameter, aggregate morphology, and the OC/TC ratio are the main descriptive parameters. However, in other studies involving post-generation treatments of particles (e.g., for coating formation, see Salo et al., 2024), the OC/TC ratio alone is indeed insufficient to fully characterize the aerosol.

Moreover, filter loading effects can differ between the two wavelengths used in the MA300. Therefore, the conclusion that one can simply select the wavelength that yields a result within ±10% of the gravimetric mass concentration is methodologically flawed and inappropriate for a journal such as Atmospheric Research.

The conclusion seems a little harsh to us; however, the initial point raised is indeed relevant and warrants further discussion. We will therefore expand upon this aspect of "filter loading effects" and the way it is addressed trough the "dual spot" approach, in the revised version of the manuscript. The aim is to provide a clearer justification of our scientific approach. Indeed, we do not assert a definitive conclusion regarding the selection of a single wavelength. The original objective was to evaluate to what extent the mass concentration estimated at 880 nm (typically associated with black carbon) can be considered representative of the total mass concentration of a carbonaceous aerosol, particularly in cases where the organic fraction varies significantly. We proposed a tentative quantitative threshold for this representativeness, but indeed this may be subject to further discussion.

The authors should provide a detailed description of the parameters used in their analysis, including the multiple scattering parameter, loading compensation parameter, and the mass absorption cross-section. Furthermore, a thorough comparison of different mass concentrations at a constant OC/TC ratio should be conducted and clearly presented.

We fully understand your request, although, for the sake of consistency with the presentation of other instruments, we had initially chosen not to include such a level of information. This additional data will be incorporated into the revised version of the manuscript. Concerning the recommendation to carry out variation of the mass concentrations at a constant OC/TC ratio, this involves changing the dilution factor for different OC/TC measurement points. So, we suggest proposing this direction in the study perspectives

Absorption at 880 nm can serve as a reliable indicator of black or elemental carbon mass across varying OC/TC ratios. In contrast, the signal in the UV range is expected to increase primarily due to enhanced scattering at higher OC/TC ratios.

This is indeed the conclusion we also arrived at, and we have attempted to go a step further by proposing a quantitative threshold, even though we acknowledge that the conclusion remains preliminary. In this study, we do have an objective measurement of the OC/TC ratio based on filter sampling and thermo-optical analysis; however, we do not currently have a methodology to quantify the organic fraction directly from the UV signal of the aethalometer. Thank you once again for providing a detailed review.

---

## Referee Report (RR2)

**Review: Evaluation of mass measurement techniques for soot with different size distributions and OC/TC contents**

After a long process, including several exchanges of comments between the authors and the reviewers, I believe that the manuscript has reached a sufficiently high level to be accepted after minor revision. My specific comments are as follows:

General:

I agree that the gravimetric measurement on filters is important in order to comply with Standard 16450. However, in the description of the experiment as well as in Figure 5, it should be clearly stated why the comparison between the gravimetric method and the TEOM is performed at all (i.e., the standard method as the reference, and on the other hand the verification of the diluter performance and the determination of losses along the different branches). It should also be clearly explained why, in your opinion, heating the sampling line to 180 °C does not significantly influence the mass measurements in either branch of the experiment. I did not find this explanation anywhere in the manuscript.

Abstract:

The abstract contains many abbreviations; some are explained in brackets, while others are not. Please minimize the use of abbreviations in the abstract, and provide their definitions in the Introduction.

Instead of listing the names of all instruments, I suggest using a more general sentence such as:

"...we assessed the mass concentration using four different online instruments, including the gravimetric method, the aerosol electrical charging method, the filter photometry method, and the aerosol mobility method."

The full names of the instruments should be presented in the Introduction.

Line 8:

Replace "OC/TC" with "organic-to-total-carbon (OC/TC) ratios "

Line 10:

Replace "The OC/TC ratio was also determined" with "The OC/TC ratio was determined by the thermal-optical method."

Figures:

Figure 2 is not referenced anywhere in the manuscript text. Please add an explicit in-text reference.

Figure 3 presents a schematic illustration of the operating principle of a filter photometer. However, it does not depict the dual-spot loading compensation principle, although the text states that "the MA300 aethalometer employs the dual-spot technology."

This inconsistency should be resolved in one of the following ways:

Update the figure caption: Modify the caption to explicitly state that the figure shows only the general operating principle and not the dual-spot technology.

Update the schematic: Adapt Figure 3 so that it correctly illustrates the dual-spot concept, ensuring consistency between the explanation in the main text and the visual representation.

Figure 8 should be split into panels (a) and (b).

Both subfigures must be referenced individually in the main text. In Figure 8(b), the grey triangles representing PPS corrected data are not visible at lower mass concentrations. The symbol style should be adjusted.

Lines: 308-311:

The paragraph 308–311 is written imprecisely, as the phrase "becomes reliable" is misleading. What you are referring to in this paragraph is that the $\sigma_{ATN}$ correction becomes linear in the UV range below 1.3 L/min (you should propose a correction factor for $\sigma_{ATN}$ (UV, <1.3 L/min)), and in the IR range between 1.3-1.5 L/min (correction factor for $\sigma_{ATN}$ (IR, 1.3-1.5 L/min). The deviation from linearity may be a consequence of changes in the MAC value as well as the multiple-scattering parameter.